# Microbial Succession under Freeze–Thaw Events and Its Potential for Hydrocarbon Degradation in Nutrient-Amended Antarctic Soil

**DOI:** 10.3390/microorganisms9030609

**Published:** 2021-03-16

**Authors:** Hugo Emiliano de Jesus, Renato S. Carreira, Simone S. M. Paiva, Carlos Massone, Alex Enrich-Prast, Raquel S. Peixoto, Jorge L. Mazza Rodrigues, Charles K. Lee, Craig Cary, Alexandre S. Rosado

**Affiliations:** 1Laboratory of Molecular Microbial Ecology, Institute of Microbiology Paulo de Góes (IMPG), Federal University of Rio de Janeiro, Rio de Janeiro 21941-901, Brazil; hugoemil@gmail.com (H.E.d.J.); ninaluacissa@yahoo.com.br (S.S.M.P.); raquel.peixoto@kaust.edu.sa (R.S.P.); 2Laboratory of Marine and Environmental Studies, Department of Chemistry, PUC-Rio, Rio de Janeiro 22541-041, Brazil; carreira@puc-rio.br (R.S.C.); cgmassone@gmail.com (C.M.); 3Multiuser Unit of Environmental Analysis, Institute of Biology, Federal University of Rio de Janeiro, Rio de Janeiro 21941-902, Brazil; aenrichprast@gmail.com; 4Department of Thematic Studies—Environmental Change, Linköping University, 58183 Linköping, Sweden; 5Red Sea Research Center (RSRC), Division of Biological and Environmental Science and Engineering (BESE), King Abdullah University of Science and Technology (KAUST), Thuwal 23955, Saudi Arabia; 6Department of Land, Air, and Water Resources, University of California, Davis, CA 95616, USA; jmrodrigues@ucdavis.edu; 7ICTAR—The International Centre for Terrestrial Antarctic Research, University of Waikato, 3216 Hamilton, New Zealand; charles.lee@waikato.ac.nz (C.K.L.); caryc@waikato.ac.nz (C.C.); 8BESE, Biological and Environmental Sciences and Engineering Division, KAUST, King Abdullah University of Science and Technology, Thuwal 23955-6900, Saudi Arabia

**Keywords:** Antarctica, hydrocarbon degradation, bioremediation, freeze–thaw, soil

## Abstract

The polar regions have relatively low richness and diversity of plants and animals, and the basis of the entire ecological chain is supported by microbial diversity. In these regions, understanding the microbial response against environmental factors and anthropogenic disturbances is essential to understand patterns better, prevent isolated events, and apply biotechnology strategies. The Antarctic continent has been increasingly affected by anthropogenic contamination, and its constant temperature fluctuations limit the application of clean recovery strategies, such as bioremediation. We evaluated the bacterial response in oil-contaminated soil through a nutrient-amended microcosm experiment using two temperature regimes: (i) 4 °C and (ii) a freeze–thaw cycle (FTC) alternating between −20 and 4 °C. Bacterial taxa, such as *Myxococcales*, *Chitinophagaceae*, and *Acidimicrobiales,* were strongly related to the FTC. *Rhodococcus* was positively related to contaminated soils and further stimulated under FTC conditions. Additionally, the nutrient-amended treatment under the FTC regime enhanced bacterial groups with known biodegradation potential and was efficient in removing hydrocarbons of diesel oil. The experimental design, rates of bacterial succession, and level of hydrocarbon transformation can be considered as a baseline for further studies aimed at improving bioremediation strategies in environments affected by FTC regimes.

## 1. Introduction

Antarctic soils are subjected to extreme environmental conditions, based on their topography and latitude and the influence of a few animal species. In general, soils are affected by long light/dark periods, high UV radiation levels, low nutrient content, strong winds, low precipitation and temperatures, and humidity fluctuations and freeze–thaw cycles [1,2,3,4,5]. These features are responsible for the presence of a drastically simplified food web [6,7], in which only highly adapted organisms are able to establish themselves, adapt, and persist [8]. The majority of these are microorganisms, comprising most of the biomass in Antarctic ecosystems [9].

Apart from Antarctic environmental conditions, microbial communities are subjected to additional selective pressure based on the growing human activity on the continent, which increases the risk of accidental oil spills on soil [10]. Oil decreases the abundance of available nutrients in Antarctic soil, strongly compromising the natural microbial attenuation and consequently affecting the balance of carbon, nitrogen, and phosphorus (C–N–P) needed to allow natural attenuation to take place [11]. Consequently, after an oil spill event, the contaminant is either not degraded or degraded very slowly, persisting in the environment for a considerable period [12,13]. Oil can also eventually be spread across soils by seasonal ice melt or rain, penetrating deeper soil layers and entering the ocean or nearby lakes [10].

Previous studies of microbial communities in contaminated Antarctic soils found that these communities have high microbial richness and diversity, showing good potential for bioremediation strategies [14,15,16,17]. However, subzero temperatures make bioremediation treatments difficult, and summer, when temperatures and microbial metabolic activity levels are higher, is the best time of the year to perform in situ manipulations [18,19,20]. However, due to the short summers and generally high contamination levels, summer periods are not long enough to achieve efficient biodegradation ratios [21,22]. Ferguson and colleagues [21] evaluated the mineralization of ^14^C-labeled octadecane in the Antarctic and concluded that despite the low degradation level, a bioremediation approach would still be sufficient for significant mitigation of oil during the summer.

A few studies have suggested the possibility of extending bioremediation treatments to cyclic freezing conditions (subjected to freeze–thaw cycles (FTCs)) [23]. Recently, McWatters and colleagues [24] performed the first complete successful large-scale in situ diesel bioremediation treatment (biopile) in Antarctica. This five-year study indicates that bioremediation approaches can be used in situ over several local freeze–thaw cycles, which confirms that biodegradation still occurs under subzero temperatures [25]. Apparently, either freezing or defrosting temperatures seem to have a significant impact on microbial metabolic activity, as found by Chang and colleagues [26], who showed that 32% of the total removal of 52% (for F2 and F3 hydrocarbon fractions) was achieved in the temperature range of 2 to −5 °C. In addition, it was shown that in permafrost soils, certain bacterial groups can only synthesize DNA at temperatures between −9 and −20 °C, but not above −6 °C [27].

Although the potential for bioremediation has been demonstrated in soils under freezing conditions [17,24,26,28], some important questions remain unanswered, such as (i) How do different microbial communities respond to nutrient-amended treatment under FTC and conventional unfrozen treatments? (ii) What does this pattern of microbial succession look like? (iii) Are different concentrations of hydrocarbons correlated with shifts in the microbial community under FTC regimes?

This study addressed the above questions for the first time. We examined and compared the bacterial community structure in soils with diesel oil contamination and nutrient amendments in different temperature regimes. These analyses were complemented by a prediction of the functional profiles of the soils and 16S rRNA transcript quantification.

## 2. Materials and Methods

### 2.1. Soil Sampling

The soil used in this study was collected from the central part of King George Island (Keller Peninsula) in the South Shetland Islands archipelago off the Antarctic Peninsula (61° S–63°30′ S and 53°55′ W–62°50′ W). The general physicochemical characteristics of the soil used in the experiment were as follows: pH: 8.27 (±0.22); total nitrogen (mg dm^−3^): not detected; total available phosphorus (mg dm^−3^): 179 (±19); potassium (mg dm^−3^): 197 (±11); iron (mg dm^−3^): 126 (±24.5); the values in parenthesis represent the standard deviation between triplicates. Detailed characteristics of the soil, such as formation process and external influences, can be found in [29].

Soil was collected during the austral summer of 2011–2012. Approximately 15 kg of soil was collected (0–15 cm deep) from 4 points at least 3 m distant from each other. The soil temperature at the sampling moment ranged from 0 to 5 °C. The soil was then sieved to remove large stones and mixed extensively for 2 h in sterilized plastic boxes with the aid of sterilized shovels. The soil was stored at −80 °C and transported to the laboratory.

### 2.2. Experimental Microcosms

The microcosm experiment was assembled by placing 50 g of soil in a glass jar (100 mL), aiming to create 3 compositions: (i) control (Ctr), containing only soil; (ii) nutrient-amended (BS), containing soil + diesel oil + fertilizer; and (iii) diesel oil (Oil), containing soil + diesel oil. Diesel oil (0.5% *v*/*v*) and fertilizer (250 mg N/kg commercial mono-ammonium phosphate (MAP), as used in a previous study [30]) were added in the laboratory to the respective soil treatments, and all 3 treatments were mixed for 2 h. The treatments were assembled in triplicate, considering 5 destructive samplings. A total of 90 glass jars containing respective treatments (30 jars for Ctr, 30 jars for BS, and 30 jars for Oil) were generated at the end of the experiment. The oxygen level was monitored, and all treatments were maintained with oxygen levels between 17 and 20% by opening the glass jars once a day for 10 min.

The experiment was started by subjecting all samples to initial acclimation at 4 °C for 15 days. After this step, samples were divided into 2 equal groups: (i) half the samples (15 glass jars from each treatment, 45 jars in total) were maintained at 4 °C throughout the experimental period (75 days) and (ii) the other half were submitted to freeze–thaw cycle (FTC) regimes, in which the temperature alternated between 4 and −20 °C every 15 days. On the last day of each 15-day period, destructive samples were collected and analyzed.

### 2.3. Hydrocarbon Quantification

Hydrocarbons were determined according to [31]. Briefly, approximately 10.0 ± 0.01 g of sediment was placed in an ultrasonic bath with dichloromethane and *p-*terphenyl-d_14_ (100 ng) as surrogate standards. Bulk extract cleanup and fractionation were performed by adsorption chromatography in a glass column packed with activated copper, anhydrous Na_2_SO_4_, alumina, and silica. Two fractions (F1 and F2) were recovered, and the total petroleum hydrocarbon (TPH) content in part of each fraction was determined with a gas chromatograph fitted with a flame ionization detector (Finnigan Focus GC-FID). The F2 fraction was analyzed for polycyclic aromatic hydrocarbon (PAH) content by gas chromatography/mass spectrometry (GC/MS) in a Finnigan Trace-ITQ 9000 system (ThermoFisher Scientific, Waltham, MA, USA). A total of 37 PAHs (Appendix A), including parental and alkylated homologs, were identified and quantified.

Hydrocarbons were quantified using the internal standard method (deuterated n-C_30_ for TPH, and a mixture of naphthalene-d_8_, acenaphthene-d_10_, phenanthrene-d_10_, chrysene-d_12_, and perylene-d_12_ for PAHs). Blank samples were extracted as part of the analytical control, and precision and accuracy were verified with certified sediment material (SRM 1944). The limit of quantification for the method was 0.010 µg g^–1^ for TPH and 0.20 ng g^–1^ for PAHs; these values were calculated based on the lowest point in the calibration curve and the mass of sediment extracted.

### 2.4. RNA Extraction and Conversion to cDNA

Soil samples were placed in an RNAlater™ (Ambion, Foster City, CA, USA) and stored according to the manufacturer’s protocol (−80 °C) until RNA extraction. All solutions, glassware, and plastics were either certified RNase-free or treated with 0.1% diethyl pyrocarbonate overnight and autoclaved. Total RNA was extracted from all samples using the RNeasy PowerSoil Total RNA Kit (Qiagen, Carlsbad, CA, USA) and quantified using a Qubit Fluorometer (Invitrogen, Carlsbad, CA, USA). First-strand synthesis of cDNA from the resulting antisense RNA was carried out with the SuperScript™ III First-Strand Synthesis System (Invitrogen, Carlsbad, CA, USA). The SuperScript Double-Stranded cDNA Synthesis Kit (Invitrogen) was used according to the manufacturer’s instructions to synthesize double-stranded cDNA, followed by a purification step using a QIAquick^®^ PCR Purification Kit (Qiagen, Valencia, CA, USA).

### 2.5. ARISA Community Fingerprint

Automated ribosomal intergenic spacer analysis (ARISA) [32] was used for preliminary visualization of microbial diversity and structure during the treatments and temperature regimes. The bacterial intergenic spacer region (ISR) in the rRNA operon was amplified using PCR primers ITSReub-Hex (5′-GCCAAGGCATCCACC-3′) and ITSF (5′-GTCGTAACAAGGTAGCCGTA-3′) [33]. All ARISA PCRs were run in triplicate on a Bio-Rad DNA Engine R^®^ (PTC-200) Peltier Thermal Cycler (Bio-Rad Laboratories, Hercules, CA, USA). The thermal cycling conditions were 94 °C for 5 min, followed by 30 cycles of 94 °C for 45 s, 55 °C for 1 min, and 72 °C for 2 min, with a final extension at 72 °C for 7 min. Once amplified, all triplicate PCRs were resolved on 1% agarose gel to ensure amplification.

The resulting data matrix was analyzed using Plymouth Routines in Multivariate Ecological Research (PRIMER) v6 [34]. Beta diversity was calculated using a resemblance matrix created using Bray–Curtis dissimilarity distance and then computed into a 2-dimensional ordination plot using nonmetric multidimensional scaling (NMDS).

### 2.6. Quantitative PCR

PCR assays were performed using the Rotor-Gene 6000™ (Corbett Research, Mortlake, NSW, Australia) for quantification of the total abundance of the gene encoding 16S ribosomal RNA, which was obtained from cDNA, according to the manufacturer’s instructions. The amplification reaction was performed in a volume of 25 µL containing a KAPA SYBR^®^ FAST qPCR Master Mix (2×) Kit (Kapa Biosystems, Woburn, MA, USA), 10 μM of each primer, and <20 ng of template DNA. The bacterial 16S rRNA gene was amplified using the primers 357F (5′-GRS CTA CGG GCA G-3′) and 529R (5′-AGC TGG TGC GGC CGC-3′) [35]. All amplifications were performed in triplicate. DNA samples included a standard (from a clone containing the gene coding for the 16S ribosomal RNA subunit) previously used to construct the standard curve, and water was used as a negative control. The PCR conditions were 95 °C for 3 min, followed by 40 denaturation cycles at 95 °C for 3 s, annealing at 60 °C for 20 min, and extension at 72 °C for 45 s, and a dissociation curve was generated after the final cycle to evaluate the formation of primer dimers. Standard curves were generated in triplicate, and the amplification of standards was linear over 5 orders of magnitude (i.e., 0.2 × 10^−5^–0.2 ng of DNA). The data were standardized to copies of the gene per gram of soil. Melting curves and agarose gel electrophoresis of the qPCR products were carried out to confirm the identity of the PCR products.

### 2.7. cDNA PCR and Sequencing

Partial 16S ribosomal RNA (rRNA) gene amplicons were produced using the universal primers 515f (5′-GTGCCAGCMGCCGCGGTAA-3′) and 806r (5′-GGACTACHVGGGTWTCTAAT-3′), which target the V4 variable region [36]. The forward primer consisted of 4 components: 5′-[adaptor library key (CCATCTCATCCCTGCGTGTCTCCGACTCAG) + an IonXpress barcode (from 10 to 12 nt) + barcode adaptor (GAT) + (forward primer)]-3′. The reverse primer consisted of 2 components: 5′-[adaptor library key (the same as detailed above) + (forward primer)]-3′. Reactions were carried out in a volume of 25 μL, which contained 1.5 U of Taq polymerase, 100 nM of each primer, 3 μL of 10× Buffer, 3 μL of 5 mM dNTPs, 1 μL of BSA, 3 μL of 25 mM MgCl_2_, 11.88 μL of H_2_O, and 1 ng of DNA. PCR cycling conditions were as follows: initial denaturation for 3 min at 94 °C, amplification for 30 cycles of 45 s at 94 °C, 1 min at 50 °C, and 1.5 min at 72 °C, a final extension for 10 min at 72 °C, and hold at 4 °C. PCR products (5 μL) were quantified by 1.0% agarose gel electrophoresis to confirm successful amplification.

The PCR products were quantified with Qubit™ dsDNA HS assay (Invitrogen, Waltham, MA, USA). Each sample was diluted to 26 pmol, pooled into a single tube, and amplified with PCR. Sequencing was conducted on an Ion Torrent Personal Genome Machine, using the Ion Xpress™ Template Kit (Life Technologies, Carlsbad, CA, USA) and the Ion 314 chip (Life Technologies) following the manufacturers’ protocols.

### 2.8. Processing Sequencing Data

Sequences were size-filtered with Mothur v.1.38.1 [37] and barcode/primer trimmed, dereplicated, and clustered into 97% operational taxonomic units (OTUs) using Usearch v.9.2 [38]. Chimera detection and removal steps were performed using Uchime2 [39] and the SILVA v.128 database (http://www.arb-silva.de (accessed on 14 February 2021)). Representative sequences of each cluster were assigned using Mothur v.1.38.1 (default k-mer-based approach) and the SILVA v.128 database [40], with an assignment threshold set at 80%.

Mothur [37] was used to calculate sequence distance matrices and cluster sequences into OTUs, defined at the furthest neighbor Jukes–Cantor distance of 0.03 (OTU_0.03_). Rank-abundance data were generated for each treatment, and rarefaction curves, collector curves, and population diversity indices were calculated.

Community similarity trees, principal component analysis (PCA) plots, and Venn diagrams were generated using R software [41] and specific packages: Phyloseq [40], Venn diagram [42], ggplot2 [43] and ape [44], plyr [45], and FactoMineR [46]. The percentage of similarity analysis (SIMPER) [47] was performed using PAST v. 3.22 [48]. The DNA sequences have been deposited in the National Center for Biotechnology Information (NCBI) database with BioProject ID PRJNA644876.

### 2.9. Statistical Analysis

Significant differences in the bacterial community and hydrocarbon content among the various treatments were analyzed by ANOVA, followed by Tukey’s test. Statistical significance in this analysis was defined as *p* < 0.05. Alpha diversity was estimated as OTU richness and Shannon index value.

## 3. Results

### 3.1. Microbial Community Structure

The ARISA technique was applied to all samples from both temperature conditions, unfrozen (4 °C) and freeze–thaw cycles (FTCs) (4 and −20 °C). Amplification profiles revealed the formation of different clusters in which samples were basically separated by the presence/absence of diesel oil and fertilizer and by the temperature condition applied.

Ordination of the bacterial community structure from ARISA indicated the presence of three major clusters (Figure 1): one including samples from the highly contaminated treatments, i.e., diesel oil (Oil) and nutrient-amended (BS) treatments; one including the control samples (Ctr); and one including a few samples from the BS and Ctr treatments. The core of contaminated and control samples could be visualized by the high percentage of similarity (>80%) for the respective clusters. Samples from the BS treatment were mostly plotted around the most contaminated soil samples (Oil).

Samples maintained under different temperature conditions, treatments, or collection times did not form any specific group, indicating that temperature and treatment were not significant grouping factors when analyzed for the middle of the incubation period. However, when samples from the last time point (t14, 75 days), clustering driven by the oil content, nutrient amendment, and temperature regime resulted in clusters that were clearly distinct from each other. Based on that, we decided to use only the last time point (75 days after the beginning of the experiment) for further downstream analysis.

### 3.2. Sequencing Results and Biodiversity

To identify the most significant bacterial groups related to specific treatments and/or temperature conditions, PCR amplicons of the partial bacterial 16S rRNA gene (hypervariable region V4) were sequenced, generating 5,685,725 raw sequencing reads ranging from 244 to 301 nt in mean read length. After removing chimeras, a total of 2,945,285 sequences remained clustered into 3141 different OTUs (3% dissimilarity cutoff). The rarefaction curve showed that saturation in sequencing coverage was achieved.

The number of bacterial OTUs showed that all soils had high levels of richness and diversity, which were clearly affected by the different treatments and temperature conditions (Table 1).

The number of OTUs ranged from 315 for the unfrozen nutrient-amended treatment (BS) to 1417 for the freeze–thaw control (FCtr). Alpha diversity results showed that the presence of oil decreased the number of OTUs in the FTC samples. For the unfrozen soils, the presence of fertilizer in the BS samples led to a decrease in the number of OTUs, while in the oil-treated samples, the number of OTUs increased. Chao richness showed the same pattern, while Shannon diversity indicated lower diversity for treatment frozen nutrient-amended soil (FBS). The decrease in diversity under certain treatments might be an indication of microbial composition turnover, allowing more specialized taxa to thrive. The treatment “frozen oil “(FOil), unfrozen Oil, and unfrozen BS treatments showed no significant differences in Shannon diversity.

Additionally, qPCR of the 16S rRNA gene showed that treatments under the FTC regime had higher 16S rRNA gene copy numbers than the other treatments. The FBS and FOil treatments showed higher 16S rRNA transcript copy numbers than the Ctr, FCtr, and BS treatments (Table 1).

### 3.3. Hydrocarbon Removal/Transformation

The hydrocarbon levels, as suggested by the content of TPH, total PAHs, and the 16 PAHs (Environmental Protection Agency (EPA) priority list), showed a common pattern for all samples: the hydrocarbon level increased from the control to the oil samples and decreased from the amended nutrient compared to oil-contaminated soil. This general pattern was observed in both unfrozen and FTC samples (Table 1).

When analyzing the same treatments but under different temperature conditions, it was possible to observe the following: (i) FCtr showed a reduced hydrocarbon level compared to the similar Ctr treatment (TPH −28% (*p* = 1.024), PAHtot −29% (*p* = 0.999), 16PAH −35% (*p* = 0.996)). This suggests a better natural attenuation affect for those soils with low contamination levels. (ii) Attenuation for unfrozen nutrient amended soil (BS) was more efficient than the similar FBS (TPH −33% (*p* = 0.891), PAHtot −29% (*p* = 0.076), 16PAH −35% (*p* = 0.619)). (iii) Natural attenuation for oiled samples under unfrozen conditions (Oil) was strong compared to the similar FOil treatment (TPH −68% (*p* = 0.344), PAHtot −49% (*p* = 0.072), 16PAH −4% (*p* = 0.997)) regarding TPH and total PAH. The results obtained for all variations of hydrocarbons are statistically significant (*p* < 0.05) between treatments, as can be observed by each sample interaction in Table 1.

### 3.4. OTU and Sample Profile Analysis

Venn diagrams were created to reveal unique and shared OTUs between the three treatments under the FTC and constant-temperature regimes (Figure 2). The shared OTU profile shows consistent differences for the same treatments in the two temperature regimes.

Overall, samples kept at 4 °C showed a higher percentage of unique OTUs in the Oil treatment (27.8%). Interestingly, 25.5% of the Oil-related OTUs were shared with the OTUs of the control samples, and 24.3% were shared among all three treatments (Figure 2A). On the other hand, both the percentage of unique OTUs from nutrient-amended treatment samples (BS) and the percentage of OTUs shared between the BS/Oil and BS/Ctr samples were less than 5%.

During the FTC regime, most OTUs (35.9%) were shared by the BS and Ctr samples, followed by 30.1% shared among all three treatments. Control samples had 20.8% unique OTUs (Figure 2B).

After assessing the bacterial profile of each sample and treatment, it was possible to identify changes that occurred in the bacterial community linked to some specific treatments. These changes were observed by a matrix generated with the variables (bacterial groups) and their frequency in each sample group (treatment). The result is shown in Table 2, where the major bacterial contributors to the treatments tested are listed, and their presence and prevalence are correlated to each treatment and temperature condition.

Based on these findings, we discovered that *Pseudomonas* and *Rhodococcus* were the most important groups, acting as the foundation of these communities in freeze–thaw conditions and at 4 °C. Additionally, these main groups were strictly linked to treatment with added diesel oil. However, their presence and dominance were still higher in biostimulation treatments (added fertilizer) for both temperature regimes evaluated.

A principal coordinate analysis (PCoA) plot was generated from the same group of samples to visualize the community distribution better. The PCoA plot shows that the community composition in the FBS samples was very similar to that in the control samples (FCtr) and was more dissimilar to the Ctr, Oil, Foil, and BS treatments (Figure 3), considering the most significant axis (Dim1, 32%). The profile was very dissimilar after BS treatment, followed by FOil treatment.

Adding the hydrocarbon content to the PCA plot revealed a strong relationship between the highest hydrocarbon levels and the BS and FOil samples (Figure 3). The FBS, Ctr, and FCtr samples had lower TPH and PAH levels than the other treatments.

By means of SIMPER analysis (similarity percentages), we correlated the bacterial species that contributed most to the profiles of the samples, based on their dissimilarity (Figure 3). Bacterial taxa, such as *Pseudomonas*, *Rhodococcus*, *Sphingobium*, *Comamonadaceae*, *Oxalobacteraceae*, *Methylobacterium*, and *Xanthomonadaceae,* were closely related to the most-contaminated samples, while *Myxococcales*, *Chitinophagaceae*, and *Acidimicrobiales* were strongly related to the FBS and FCtr samples. The detailed values found in the SIMPER analysis are shown in Table 2.

### 3.5. Taxonomy

Taxonomic analysis of bacterial composition at the phylum level revealed differences among treatments and between temperature regimes. Of the 39 phyla identified in all samples, the most abundant phylum was Proteobacteria (52.6%), followed by Actinobacteria (22.3%), Bacteroidetes (9.7%), Firmicutes (5.3%), Chloroflexi (2.3%), Acidobacteria (2.2%), Gemmatimonadetes (2.1%), Planctomycetes (1.5%), and Verrucomicrobia (1.0%). Proteobacteria and Actinobacteria were the most abundant in all treatments, ranging from 44.08% in the FCtr sample to 60.86% in the BS sample. The abundance of Actinobacteria ranged from 18.58 to 25.81% in the Oil and FOil samples. Bacteroidetes (4.88%) was the least abundant, and Firmicutes (6.10%) was the most abundant in the BS treatment. Additionally, some phyla were present at lower than 1% abundance only in the BS treatment, while the others showed abundances ranging from 1 to 3%; these were Chloroflexi, Acidobacteria, Gemmatimonadetes, Planctomycetes, and Verrucomicrobia (Figure 4).

The response to treatment was also marked at the genus level. *Rhodococcus* was positively related to contaminated soils but was further stimulated under FTC conditions (Figure 4B). In contrast, although the genus *Pseudomonas* has also been associated with contaminated soils, it was only upregulated in the FBS treatment. Indeed, both *Rhodococcus* and *Pseudomonas* were most abundant in the FBS treatment. The relative abundance of the genus HB2−32−21 decreased in the presence of diesel oil, while the relative abundance of *Methylobacterium* increased in the unfrozen nutrient-amended samples (BS) (Figure 4B).

## 4. Discussion

Until recently, bioremediation treatment of polar soils was known to be successful during the summer periods, when temperatures are above the freezing point, and consequently, the microbial activity is higher [15,26]. However, due to the short summer in these regions and often-missed remediation goals, further studies on the possibility of using winter periods to optimize bioremediation techniques have been encouraged.

Previous studies have shown that in arctic soils, the hydrocarbon-degrading microbial community is not eliminated under subzero temperatures [26,49,50] and that a highly diverse array of bacteria are active even at −20 °C [27,51]. This also seems to be true for the Antarctic soil microbial community, based on our findings of a very specific bacterial profile (Figure 1). Moreover, oiled samples under the FTC regime (FOil) showed very slight hydrocarbon removal compared to similar unfrozen samples (Table 1), which underscores the limitations of natural removal in the Antarctic environment and the importance of studying microbial succession in environments affected by FTC regimes.

The comparison of oil removal rates between oiled samples (Oil and FOil) and nutrient-amended samples (BS and FBS) showed that oil degradation advanced further under FTC conditions more intensely than under similar unfrozen treatment conditions (Oil and BS). This suggests that nutrient amendment has a greater effect in freeze–thaw than unfrozen conditions because the features of these environments in the frozen state cause a long delay in the natural degradation of contaminants. On the other hand, unfrozen soil showed higher oil removal rates for both oiled (Oil) and nutrient-amended (BS) samples, which probably reflects a faster removal process under warmer conditions (above zero temperature). For locations where it is not possible to attain continually unfrozen conditions, the nutrient-amended treatment seems to be a strategy that would more efficiently favor the development of a higher diversity of hydrocarbon-degrading microorganisms. However, it is important to highlight that the observed removal of oil may not be completely due to its biodegradation since some abiotic processes, such as volatilization, physical separation, and partitioning with soil phases, may also influence the removal of oil in situ [50]. Measuring these additional variations in further experiments could elucidate the budget of oil removal during FTC experiments.

Our results showed differences in the relative abundance of bacterial phyla and genera well known for containing notable hydrocarbon-degrading species, such as *Pseudomonas*, *Sphingobium*, and *Rhodococcus* [52,53]. According to our findings, the genera *Rhodococcus* and *Pseudomonas* were the most abundant in all samples, especially in oil-contaminated treatments, which hosted the highest abundance of these bacteria. The presence and predominance of these alkane-degrading genera in contaminated polar soils were observed in [54], as well as by Kim and colleagues [17] in a study evaluating the effect of the FTC regime on a biopile compared to non-amended treatment. As also revealed in our study, Kim and colleagues observed shifts in the bacterial community under different seasonal phases and found the same key genera inhabiting FTC nutrient-amended soil. However, the genus *Thiobacillus* was the most abundant genus in their samples, while the abundance of this genus was less than 2% in our study. According to Kim and colleagues, high amounts of sulfur and iron in their soils could explain the high abundance of *Thiobacillus*.

In addition to the most abundant bacterial genera, other key genera were also found in our samples and were positively associated with only the FBS treatments. One was *Psychrobacter,* which is a group of psychrophilic bacteria with biotechnology potential frequently isolated from polar sites, but several strains have yet to be fully characterized [55,56,57,58]. Others were *Arenibacter*, which was recently discovered and isolated from Antarctic marine sediment [59], and shows the ability to biodegrade hydrocarbon compounds [60]; *Gemmata*, which shows the ability to oxidize ammonia [61]; *Methyloversatilis*, which contains thermotolerant species [62] and is capable of degrading benzene and naphthalene [63], an important characteristic in view of our experimental conditions; *Parvibacullum*, which is strongly associated with hydrocarbon biodegradation in nutrient-amended and non-nutrient-amended soils [62,64,65] but is also widely found in cold environments [60,66,67]; *Fluviicola*, which shows a biodegradation profile against persistent organic pollutants [68] and has a close relationship with shifts in temperature, strongly correlating with effective pollutant removal at lower temperatures [69]; *Planctomyces*, which is found in high abundance in phytoremediation experiments, and demonstrates a close correlation with the degradation of C21–C34 petroleum hydrocarbon fractions [70]; *Gemmatimonas*, which was previously associated with hydrocarbon degradation under anaerobic conditions in soils from the same location as the soils in our study [16], as well as in agricultural soils [71], and has also shown a positive correlation with fertilized soils [61]; and *Dechloromonas*, which is associated with decomposition of contaminants through denitrification [72]. Based on the above information, the microbial community established in the FBS treatment includes several genera with well-known biodegradation capabilities.

Chang and colleagues [26] demonstrated that the 16S rRNA copy number increased by the end of the experimental freezing period, after which there was no increase during the frozen state. These results agree with our data, which indicated higher potential activity, as observed by the 16S rRNA copy number (obtained from cDNA), for contaminated soils under the freeze–thaw regime (Table 1). According to recent predictions based on the quantification of 16S rRNA transcripts, we suggest that these values indicate potential activity instead of the number of individual cells, the growth rate, or the metabolic state [73]. The reason is that the number of ribosomes in a cell is linked to the potential activity limit of the cell. Thus, considering our experimental design, the quantification of the 16S rRNA gene indicated a clear environmental effect on the bacterial community, providing information about the functional potentiality of our samples. Temperature alternations seem to have had the opposite effect on the control samples (FCtr) since the 16S rRNA copy number decreased, indicating lower potential activity under the FTC regime. Surprisingly, not even the strong freezing effect, from 4 to −20 °C without a pre-freezing step, led to a decrease in the potential activity of the bacterial community in contaminated soils (FBS and FOil). These findings reveal that the bacterial population is highly resistant to intracellular ice crystallization and cell damage [49]. These results are also supported by the fact that alkane degradation genotypes are commonly detected among cold-adapted bacteria [54], which confirms the higher potential of degradation for these cold-adapted organisms.

According to the results observed for the control samples, the expected effect of the utilization of nutrients from killed cells by surviving cells after freeze–thaw periods and the disruption of soil aggregates causing the liberation of nutrients [74] seem not to have happened strongly enough to lead to a significant increase in the abundance of specific groups and/or an increase in the total microbial activity in the soil. However, it is also important to consider that: (i) the duration of the experiment may not have been long enough [23] to reveal this effect in soil that was not subjected to larger stress, as it was for nutrient-amended and/or oil treatments; (ii) microbes are carbon-limited at subzero temperatures [75]; and (iii) there is evidence that the total abundance of RNA is higher when the hydrocarbon degradation rate is highest [49]. Those previous results might explain the much higher potential activity of the treatments supplied with an external carbon source (e.g., diesel oil) resulting from translation from a high 16S rRNA copy number. This result suggests that contaminated soils might have a freezing temperature effect that increases the bacterial population up to a frozen plateau, during which the population size remains static until the next thawing period when the population increases again.

Despite the undeniable effect of different treatments, when two of the three studied factors (temperature regime, collection time point, and nutritional treatment) were analyzed together or separately, no strong and consistent effect was observed over the bacterial community. In a recent study evaluating different nutrient amendment techniques, the response of the microbial community to hydrocarbon degradation and changes in the size of the population was also significant only in frozen soils [28], suggesting a temperature-dependent microbial response. This confirms the complexity of soil microbial interactions, even for a food-web structure considered to have low complexity, such as the terrestrial Antarctic environment [76]. The concomitant contributions of different factors, such as temperature, hydrocarbon concentration, and biological activities, seem to be crucial for the efficiency of bioremediation treatment [77]. Additionally, this study shows that nutrient-amended treatment increased the presence of oil-degrading bacteria and the removal of hydrocarbons by 20%.

The FBS treatment showed a very similar OTU profile to the control samples (35.9%), while the unfrozen BS and Oil samples showed very different community structures compared to the control. The profiles of the unfrozen and FTC treatment samples also differed. The unfrozen samples showed a higher specific OTU profile for the diesel-oil treatment, while the FTC regime shared OTUs between the FBS and FCtr treatments. The other treatments presented completely different profiles, and none of them revealed a profile similar to that of the unfrozen control samples (Ctr) (see Figure 3). The application of fertilizer under an FTC regime favors the conservation of a microbial community structure similar to the soil without the addition of diesel oil or fertilizer (FCtr) and should be considered as a preferential treatment to be used in soil recovery from hydrocarbon contamination in the Antarctic, and also applied to other colder environments affected by FTC conditions.

Similar studies comparing nutrient-amended treatments under freeze–thaw cycles also indicated that there was significant microbial stimulation during the FTC regime with a decrease in F2 (C10–C16) and F3 (C16–C34) compounds according to ^14^C-hexadecane mineralization assay [26]. Hydrocarbon degradation kinetics during freeze–thaw periods also increased from 2 to 7 when nutrient-amended biopiles were applied, and the removal of F2, F3, and TPH was 57, 58, and 58%, respectively, under FTC conditions [17]. On the other hand, Atlas and Bragg [78] evaluated samples and contamination levels from the Exxon Valdez spill and did not recommend the use of bioremediation approaches when natural attenuation would be preferable. On the contrary, our study revealed that natural attenuation had no significant response, and the general contamination levels stayed high under unfrozen and freeze–thaw regimes (Table 1). Thus, in our study the degradation rate of HPAs was higher under BS and FBS treatment, with more relevance to the freeze–thaw regime.

Regarding 16S rRNA quantification, which revealed a twofold increase from unfrozen BS to FBS treatment and decreased PAH and TPH levels, we suggest that the nutrient-amended treatment under the freeze–thaw temperature regime (FBS) can be more efficient at removing hydrocarbons, and this is associated with a reduced microbial profile disturbance (Table 1 and Figure 3).

## 5. Conclusions

Although recent studies have demonstrated that microbial activity exists at subzero temperatures and that nutrient-amended processes might have a significant impact on the microbial community, and consequently on remediating the impacted environment [17,26,27,28,79,80], none of the related studies demonstrated how bacterial communities have been shaped by nutrient-amended treatment during positive temperatures (simulating the Antarctic summer) or temperature fluctuations (summer + winter), a crucial aspect when designing strategies for the recovery of impacted Antarctic soils. Thus, based on the bacterial succession patterns and oil removal observed, we suggest that nutrient amendment under the FTC regime can be a promising approach for further studies regarding applied bioremediation strategies to improve oil removal rates, with a lower impact on the soil microbial community.

## Figures and Tables

**Figure 1 microorganisms-09-00609-f001:**
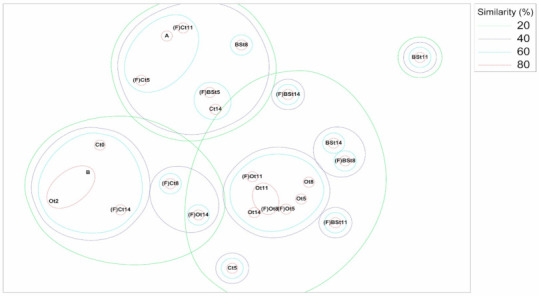
Nonmetric multidimensional scaling (NMDS) ordination of bacterial automated ribosomal intergenic spacer analysis (ARISA) community composition based on Bray–Curtis distances (stress = 0.13)). Sample IDs are represented as follows: Freeze–thaw cycle (FTC) regime (F) (if applied) + treatment (control (C); nutrient amendment (BS); oil (O)) + collection time (t). Points A and B represent samples plotted in the same site: A: BSt0, BSt2, and BSt5; B: Ct2, Ct8, Ct11, Ct14, and Ot0.

**Figure 2 microorganisms-09-00609-f002:**
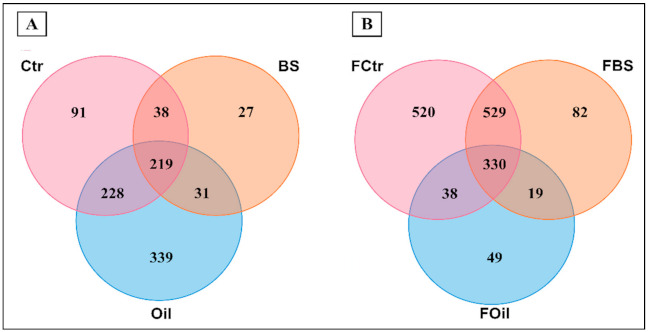
Venn diagram representing operational taxonomic units (OTUs) shared between treatments (Ctr, control; BS, nutrient amendment; Oil, diesel oil) at (**A**) 4 °C and (**B**) under the FTC regime.

**Figure 3 microorganisms-09-00609-f003:**
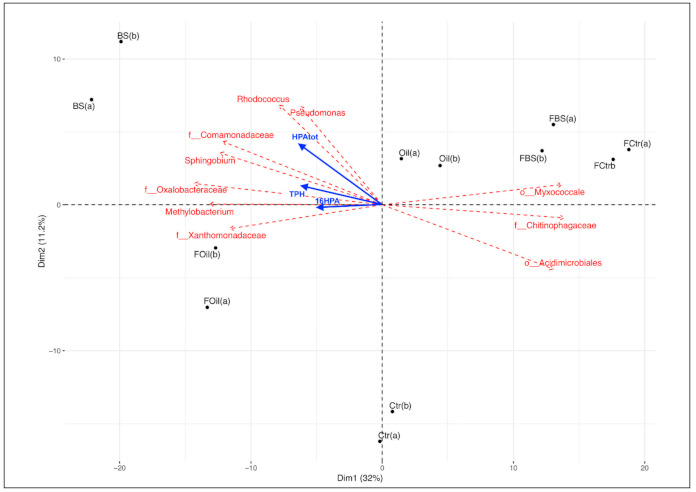
Principal coordinate analysis (PCoA) with multidimensional scaling (MDS) ordination of bacterial OTUs based on Bray–Curtis distances (stress = 6.57 × 10^−5^). BS, nutrient amendment; Ctr, control; Oil, diesel oil. F is added to the treatment code if the sample was under a freeze–thaw cycle (FTC).

**Figure 4 microorganisms-09-00609-f004:**
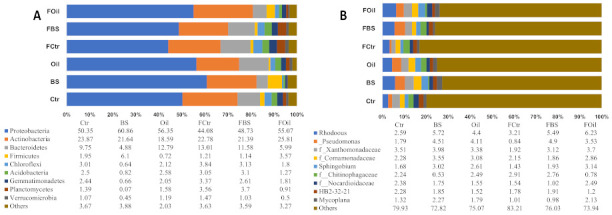
Bar graph representing relative abundance and distribution of 10 most abundant taxa at (**A**) phylum and (**B**) Operational taxonomic unit (OTU) across six treatment conditions. “Others” represent the sum of the remaining groups.

**Table 1 microorganisms-09-00609-t001:** Richness and diversity indices, real-time quantification of 16S rRNA (cDNA), and hydrocarbon quantification for all treatments after 75 days of the experiment, followed by standard deviation (±). ANOVA was significant for community structure (*p* < 0.01), total petroleum hydrocarbon (TPH) (*p* = 0.40 × 10^−3^), total polycyclic aromatic hydrocarbon (PAHtot) (*p* = 1.009 × 10^−7^), and 16PAH (*p* = 1.52 × 10^−6^). Different letters represent statistically different samples. OTU, operational taxonomic unit; FCtr, freeze–thaw control; FBS, frozen nutrient-amended soil; FOil, frozen oil.

Treatments	OTUs	Chao Estimator	Shannon Index	qPCR (Gene Copies g^−1^)	TPH (μg g^−1^)	PAHtot (μg g^−1^)	16PAH (μg g^−1^)
FCtr	1417 ± 40.31	484 ± 14.14	5.27 ± 0.12	1.2 × 10^8^	49 ± 8.50 ^a^	507 ± 137.78 ^a^	123 ± 34.56 ^a^
FBS	960 ± 83.44	431 ± 13.44	5.11 ± 0.17	1.5 × 10^4^	566 ± 38.69 ^bc^	19681 ± 248.03 ^bc^	616 ± 59.32 ^b^
FOil	436 ± 7.78	303 ± 4.31	4.86 ± 0.89	6.2 × 10^8^	1262 ± 280.05 ^b^	28667 ± 4590.36 ^b^	1370 ± 330.52 ^c^
Ctr	576 ± 24.75	364 ± 40.73	5.13 ± 0.57	1.1 × 10^3^	68 ± 24.50 ^a^	715 ± 258.08 ^a^	189 ± 39.37 ^ab^
BS	315 ± 4.95	234 ± 1.56	4.68 ± 0.34	1.1 × 10^2^	381 ± 23.64 ^ac^	13904 ± 2043.34 ^c^	398 ± 43.84 ^ab^
Oil	817 ± 35.72	372 ± 25.54	4.96 ± 0.37	4.5 × 10^2^	409 ± 167.70 ^bc^	14495 ± 2679.73 ^bc^	1310 ± 227.32 ^c^

For each column of hydrocarbon degradation (TPH, PAHtot, and 16PAH), different letters in superscript (a, b, c) indicate that the samples are significantly different, the same letters represent results not significantly different between each other (Tukey’s: *p* < 0.05).

**Table 2 microorganisms-09-00609-t002:** SIMPER results for tested treatments showing the first 10 taxa that most contributed to principal coordinate analysis (PCoA) ordination.

Temperature Condition	Taxon	Contrib. %	Cumulative %	Mean Ctr	Mean FCtr	Mean BS	Mean FBS	Mean Oil	Mean FOil
Both conditions (4 °C and freeze/thaw)	*Pseudomonas*	3.232	3.232	0.0179	0.0084	0.0451	0.0490	0.0411	0.0353
	*Rhodococcus*	2.979	6.211	0.0259	0.0321	0.0572	0.0549	0.0440	0.0623
	*Methylobacterium*	2.076	8.287	0.0128	0.0039	0.0280	0.0027	0.0012	0.0173
	*f__Chitinophagaceae*	2.009	10.3	0.0224	0.0291	0.0053	0.0276	0.0249	0.0078
	*o__Acidimicrobiales*	1.888	12.18	0.0202	0.0267	0.0020	0.0178	0.0108	0.0071
	*f__Oxalobacteraceae*	1.793	13.98	0.0137	0.0053	0.0290	0.0063	0.0111	0.0197
	*g__Sphingobium*	1.473	15.45	0.0168	0.0143	0.0302	0.0193	0.0261	0.0314
	*f__Xanthomonadaceae*	1.357	16.81	0.0351	0.0192	0.0398	0.0312	0.0338	0.0370
	*f__Comamonadaceae*	1.313	18.12	0.0228	0.0215	0.0355	0.0186	0.0308	0.0286
	*o__Myxococcales*	1.289	19.41	0.0092	0.0149	0.0020	0.0170	0.0084	0.0018
4 °C	*Rhodococcus*	3.031	3.031	0.0259	-	0.0572	-	0.0440	-
	*Pseudomonas*	2.842	5.873	0.0179	-	0.0451	-	0.0411	-
	*Methylobacterium*	2.6	8.474	0.0128	-	0.0280	-	0.0012	-
	*f__Chitinophagaceae*	1.942	10.42	0.0224	-	0.0053	-	0.0249	-
	*o__Acidimicrobiales*	1.827	12.24	0.0202	-	0.0020	-	0.0108	-
	*f__Oxalobacteraceae*	1.714	13.96	0.0137	-	0.0290	-	0.0111	-
	*g__Corynebacterium*	1.606	15.56	0.0045	-	0.0168	-	0.0000	-
	*g__Staphylococcus*	1.485	17.05	0.0059	-	0.0155	-	0.0000	-
	*Unassigned*	1.475	18.52	0.0075	-	0.0165	-	0.0010	-
	*Sphingobium*	1.369	19.89	0.0168	-	0.0302	-	0.0261	-
Freeze/Thaw	*Pseudomonas*	4.116	4.116	-	0.0084	-	0.0490	-	0.0353
	*Rhodococcus*	3.283	7.399	-	0.0321	-	0.0549	-	0.0623
	*f__Chitinophagaceae*	2.376	9.776	-	0.0291	-	0.0276	-	0.0078
	*o__Acidimicrobiales*	2.066	11.84	-	0.0267	-	0.0178	-	0.0071
	*f__Xanthomonadaceae*	1.822	13.66	-	0.0192	-	0.0312	-	0.0370
	*Sphingobium*	1.817	15.48	-	0.0143	-	0.0193	-	0.0314
	*o__Myxococcales*	1.716	17.2	-	0.0149	-	0.0170	-	0.0018
	*f__Oxalobacteraceae*	1.688	18.88	-	0.0053	-	0.0063	-	0.0197
	*g__Methylobacterium*	1.679	20.56	-	0.0039	-	0.0027	-	0.0173
	*f__Nocardioidaceae*	1.544	22.11	-	0.0154	-	0.0102	-	0.0249

Sample IDs are represented as follows: BS, nutrient amendment; Ctr, control; Oil, diesel oil. F is added to the treatment code if the sample was under a freeze–thaw cycle (FTC).

## Data Availability

All data generated or analyzed during this study are included in this published article, except for raw sequence reads that were uploaded to the National Center for Biotechnology Information’s Sequence Read Archive under BioProject ID PRJNA644876.

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
