# Peer review of "Microbial Succession under Freeze–Thaw Events and Its Potential for Hydrocarbon Degradation in Nutrient-Amended Antarctic Soil"

_microorganisms, 2021, doi:10.3390/microorganisms9030609_

Round 1

Reviewer 1 Report

The paper by Hugo Emiliano de Jesus and colleagues describes a microcosm experiment on hydrocarbon degradation by Antarctic soil microbes. An interesting topic and also relevant for bioremediation efforts. However, the execution in the Results section is severely lacking. Figure 1 – partially cut off in my copy of the manuscript, not very esthetically pleasing. What method/program was used to generate this image? Table 1 seems to be the central point of this paper, but I’m having difficulties understanding it. The concentrations of the hydrocarbons – are they the final conc. after 75 days? What was the initial concentration? What do the letters a-b-c stand for? Groups? What groups? The caption says: “Different letters represent statistically.” – I don’t know what that means. Table 2 is a complete mystery to me. Then we have Figure 4 which was also cut off in my version. Do the bars represent the values in the table beneath them? Seems like they don’t. This is all very confusing. The discussion seems ok, but without the proper presentation of the results it is very difficult to verify those findings. There are some language errors, especially in the Abstract (!). Overall the paper needs major revisions at the basics: the display of the results. You need to especially focus on the changes that occurred in those 75 days, most notably in the hydrocarbon concentration.

Author Response

Comments to Reviewer 1 (in red):

The paper by Hugo Emiliano de Jesus and colleagues describes a microcosm experiment on hydrocarbon degradation by Antarctic soil microbes. An interesting topic and also relevant for bioremediation efforts. However, the execution in the Results section is severely lacking.

#Figure 1 – partially cut off in my copy of the manuscript, not very esthetically pleasing. What method/program was used to generate this image?

We are sorry for the problem uploading the figure in our first version. The figure is correctly uploaded in our revised version.

This result was obtained by using the program PRIMER v6 (Plymouth) [1]. Beta diversity was calculated using a resemblance matrix created using Bray-Curtis dissimilarity distance and then computed into a two-dimensional ordination plot using nonmetric multidimensional scaling (NMDS).

This information is clarified in section 2.5 of methodology, between lines 168 and 171

#Table 1 seems to be the central point of this paper, but I’m having difficulties understanding it. The concentrations of the hydrocarbons – are they the final conc. after 75 days? What was the initial concentration? What do the letters a-b-c stand for? Groups? What groups? The caption says: “Different letters represent statistically.” – I don’t know what that means.

1 – The concentrations of the hydrocarbons – are they the final conc. after 75 days?

Yes. We now have included this information in the legend = the results in table 1 are referred to the samples after 75 days of experiment.

2 - What was the initial concentration?

The control soils (FCtr and Ctr) are the same then the soils with addition of diesel oil (0.5% v/v) and fertilizer (250 mg N/kg commercial mono-ammonium phosphate - MAP)). So the purpose is to compare the concentration of the control soil with the treatments after 75 days of experiment, and thus disregarding the effect of natural attenuation.

3 - What do the letters a-b-c stand for?

These letters mean “significant difference between samples”. Same letters in different samples means that those samples are not significantly different to each other. Based on this misunderstanding the explanation for this table was reformulated as follow:

Significant difference (a, b, c), Tukey’s – p < 0.05. Same letters appear when values are not significantly different between two or more treatments.” (lines 276 and 277)

4 - Groups? What groups?

This term was changed to “treatment”

5 - “Different letters represent statistically.” – I don’t know what that means.

This phrase was edited to better explain and now it is:

“Different letters represent samples statistically different.” (lines 274 and 275)

#Table 2 is a complete mystery to me. Then we have Figure 4 which was also cut off in my version. Do the bars represent the values in the table beneath them? Seems like they don’t. This is all very confusing. The discussion seems ok, but without the proper presentation of the results it is very difficult to verify those findings. There are some language errors, especially in the Abstract (!). Overall the paper needs major revisions at the basics: the display of the results. You need to especially focus on the changes that occurred in those 75 days, most notably in the hydrocarbon concentration.

1 - Table 2 is a complete mystery to me.

This table shows the similarity percentages (SIMPER) procedure (Clarke, 1993). It attempts to assess the contribution of each variable analyzed in a study by using a Bray-Curtis dissimilarity matrix. It tries to identify possible variables which are likely the major contributors to difference between groups of samples.

Clarke KR (1993) Non-parametric multivariate analyses of changes in community structure. Austral Ecol. 18:117–143.

2 – “Then we have Figure 4 which was also cut off in my version. Do the bars represent the values in the table beneath them? Seems like they don’t. This is all very confusing.”

We are sorry for the problem in our first version. The Figure 4 was improved and uploaded correctly in our revised version. 

Figure 4 is in the Taxonomy section 3.5. It represents how the bacterial community is distributed in all treatments and temperature conditions. The figure also shows the relative abundance of bacterial phyla in the samples and explains the distributions of these communities. 

General comments:

The authors agree with the reviewer that the order of table 2 and the lack of a more detailed explanation of these results led to confusion and misunderstanding. With that in mind, the authors made changes that seek to clarify the ideas. The changes were as follows:

The reference by Clarke KR (1993) was included in line 224 on the use of SIMPER analysis.
Two paragraphs were included between lines 332 and 344 explaining the purpose and results present in Table 2.
Table 2 was placed just below the two new paragraphs mentioned above.
As Figure 3 was made with the information obtained in tables 1 and 2, it is now located after table 2.

We would like to thank the reviewer for the valuable and important comments.

Reviewer 2 Report

The manuscript by de Jesus et al. studied the microbial succession under freeze-thaw events and its potential for hydrocarbon degradation. These results are of potential interest to a broad audience, specifically those involved in developing bioremediation technology and interested in microbial interactions in arctic environments. The paper is generally well-structured and the experiment was well-designed. Overall this manuscript will be acceptable after the below edits:

  1. Line 109 – please precise – does the glass jars were open or closed? If closed – what about the oxygen level during the experiment? Hydrocarbon degradation is more effective in optimum oxygen conditions. 
  2. Hydrocarbon quantification section – do the Authors checked the phenomenon of spontaneous volatilization of hydrocarbons during the experiment? 
  3. I think that in line 127 the “±” sign is missing
  4. Line 178 – I think that the “µ” sign is missing (should be 10 µM, not 10 M)
  5. Line 187 – the “-5” should be in superscript
  6. Line 187-188 – How the Authors standardized the data to gene content per gram of soil? Please specify. 
  7. Line 199-200 – please specify the amount of used polymerase in unit (U) and the number of primers used in µM. 
  8. Figure 4 – the picture is cut, I cannot see part B in .pdf version of the manuscript. Moreover, the letters are too small. 
  9. Please italicize family names of bacteria e.g. in lines 342-343
  10. Conclusions – Please reflect on the achieved aims.

Author Response

Author's reply to Reviewer 2 (in red):

The manuscript by de Jesus et al. studied the microbial succession under freeze-thaw events and its potential for hydrocarbon degradation. These results are of potential interest to a broad audience, specifically those involved in developing bioremediation technology and interested in microbial interactions in arctic environments. The paper is generally well-structured and the experiment was well-designed. Overall this manuscript will be acceptable after the below edits:

The authors wish to thank the reviewer and are glad to know that he/she have enjoyed the manuscript. 

#Line 109 – please precise – does the glass jars were open or closed? If closed – what about the oxygen level during the experiment? Hydrocarbon degradation is more effective in optimum oxygen conditions.

During the experiment the oxygen level was monitored and all treatments were kept with oxygen levels between 17 and 20% by opening the glass jars once a day for 10 minutes.

The procedure mentioned above is now included in the manuscript (lines 119 and 120)

#Hydrocarbon quantification section – do the Authors checked the phenomenon of spontaneous volatilization of hydrocarbons during the experiment?

Accuracy of the method was verified by the successful analysis of the Standard Reference Material (SRM) 1944, New York/New Jersey Waterway Sediment, from NIST. In addition, QA/QC procedures also involve the analysis of laboratory blanks (one in each batch of 10 samples), recovery of surrogate standards (acceptable between 60 and 120%) and precision estimate tests (better than 20%)

#I think that in line 127 the “±” sign is missing

The sign was included

#Line 178 – I think that the “µ” sign is missing (should be 10 µM, not 10 M)

This sign was edited to 10 µM

Line 187 – the “-5” should be in superscript

The “-5”was edited and now it is in superscript

Line 187-188 – How the Authors standardized the data to gene content per gram of soil? Please specify.

The gene content per gram of soil was achieved based on the previously construction of the standard curve (from a clone containing the gene coding for the 16S ribosomal RNA subunit). Based on the knowledge about how many gene copies we had in the different dilutions from the standard curve (by checking the fluorescence) is possible to correlate the fluorescence emitted from the sample (in which we already know the volume of soil that was used) and determine the gene copies by gram of soil.

#Line 199-200 – please specify the amount of used polymerase in unit (U) and the number of primers used in µM.

This information was included in lines 199 and 200, new lines 206 and 207

#Figure 4 – the picture is cut, I cannot see part B in .pdf version of the manuscript. Moreover, the letters are too small.

The picture was included in a complete format and the letters are now bigger

#Please italicize family names of bacteria e.g. in lines 342-343

These names were edited and now are in italic format

Conclusions – Please reflect on the achieved aims.

We edited the conclusions and the English was improved to make it clearer and the text flowed better.

References checked:

  1. Clarke, K.R.; Gorley, R.N. Primer V6: User Manual/Tutorial; Plymouth, UK, 2006;
  2. de Jesus, H.E.; Peixoto, R.S.; Cury, J.C.; van Elsas, J.D.; Rosado, A.S. Evaluation of soil bioremediation techniques in an aged diesel spill at the Antarctic Peninsula. Applied microbiology and biotechnology 2015, doi:10.1007/s00253-015-6919-0.

Reviewer 3 Report

Dear Editor,

in the paper entitled “microbial succession under freeze-thaw events and its potential for hydrocarbon degradation in a nutrient-amended Antarctic soil” the Authors assessed the microbial population and hydrocarbon degradation in three different microcosms incubated at 4 ° C or subjected to cycles of freezing (-20 ° C) and thawing (4 ° C). The results are providing interesting and valuable information. However, there are some problems and flaws in the results section.

 Major comments:

 Pag.3 Line 94. What is the soil temperature of King George Island during the summer and winter? The Authors should be taking it to account that -20°C could be a very harsh-condition for microbial growth. The process of microbial growth and biodegradation could be more efficient at temperatures similar to physiological ones.

Pag. 3 Line 112. It is not clear why the Authors used fertilizer. In addition, the effects of fertilizer on microbial community are poorly described.

Pag. 5 Line 233. Paragraph 3.1 is confusing, and the purpose is elusive. A brief introduction is useful for the reader to understand the results of this paragraph. Furthermore, it is not clear on which samples and when the ARISA analysis was performed. What is the composition of the Antarctic soil before treatments? What do the authors mean with different temperature conditions? In Figure 1 the temperature conditions are not mentioned.

Pag. 6 Line 256. Also, in this paragraph the purpose is elusive, and it is unclear when the Authors performed the 16 rRNA sequencing. Please clarify. If section 3.2 is the introduction of the following, I suggest integrating this part into section 3.3.

Table 1. It is very difficult to understand the quantification data of the biodegradation of hydrocarbons. A simple way to represent these data is through three histograms in which, for each condition (ctr, oil and BS), the quantity of hydrocarbon at time 0 and after treatment at different temperatures are reported.

Pag. 12 Line 377. In the discussion, the Authors should consider that protists are component of the soil microbiome (see for instance https://doi.org/10.1093/femsre/fuy006) and some of these are able to live in extreme environments (see for instance https://doi.org/10.1016/j.ejop.2020.125720). It would be interesting to evaluate both the role of bacteria and that of protists in the decontamination of Antarctic soils. 

The Authors used a lot of abbreviations that make the text difficult to read and understand.

Minor comments:

Title: amendeded should be amended.

Pag. 2 Line 96. Please clarify how the Authors determine the physicochemical composition of the soil.

Pag.3 Line 124. Please clarify why in the freeze and thaw cycles the Authors alternated the temperature every 15 days. Is it a standard protocol?

Pag 3. Line 127 It is unclear the amount of soil the Authors used. Please clarify.

Pag.5 Line 210. Please add city and state.

Figure 1. Figure resolution is too low.

Figure 4. This figure is incomplete.

Author Response

Author's reply to Reviewer 3 (in red):

#in the paper entitled “microbial succession under freeze-thaw events and its potential for hydrocarbon degradation in a nutrient-amended Antarctic soil” the Authors assessed the microbial population and hydrocarbon degradation in three different microcosms incubated at 4 ° C or subjected to cycles of freezing (-20 ° C) and thawing (4 ° C). The results are providing interesting and valuable information. However, there are some problems and flaws in the results section.

Major comments:

#Pag.3 Line 94. What is the soil temperature of King George Island during the summer and winter? The Authors should be taking it to account that -20°C could be a very harsh-condition for microbial growth. The process of microbial growth and biodegradation could be more efficient at temperatures similar to physiological ones.

As explained in line 104: "The soil temperature at the time of sampling was varying from 0 to 5 ° C."

It is important to note that the use of -20 ° C was an attempt to approach the winter condition on the soil of King George Island. The authors are aware of this severe condition. However, the objective here was to observe the effect of the defrost using a temperature that, despite being severe, is not unrealistic. In addition, the results showed that, although severe, the bacterial communities were still viable, even after two freeze / thaw cycles.

#Pag. 3 Line 112. It is not clear why the Authors used fertilizer. In addition, the effects of fertilizer on microbial community are poorly described.

The fertilizer was used and in this concentration due to a previous study by the same authors [Jesus et al., 2015] that revealed a result of efficient degradation by the use of similar soils. Now, the goal was to see if it would also be effective in these temperature conditions.

PS: the previous study mentioned was included in line 113

* Jesus, H.E.; Peixoto, R.S.; Cury, J.C.; van Elsas, J.D.; Rosado, A.S. Evaluation of soil bioremediation techniques in an aged diesel spill at the Antarctic Peninsula. Applied microbiology and biotechnology 2015, doi:10.1007/s00253-015-6919-0.

#Pag. 5 Line 233. Paragraph 3.1 is confusing, and the purpose is elusive. A brief introduction is useful for the reader to understand the results of this paragraph. Furthermore, it is not clear on which samples and when the ARISA analysis was performed. What is the composition of the Antarctic soil before treatments? What do the authors mean with different temperature conditions? In Figure 1 the temperature conditions are not mentioned.

We agree with the reviewer's suggestion and decided to include a brief explanation of the samples used in the ARISA analyzes, as well as the temperature regimes mentioned, lines 239 to 243.

#Pag. 6 Line 256. Also, in this paragraph the purpose is elusive, and it is unclear when the Authors performed the 16 rRNA sequencing. Please clarify. If section 3.2 is the introduction of the following, I suggest integrating this part into section 3.3.

The authors agree with the point raised by the reviewer that the text was not in an appropriate order and made the following changes:
The biodiversity results, which were in section 3.3, have been moved to section 3.2 and their section name has been edited to “3.2 Sequencing results and Biodiversity”.

Section 3.3 has been renamed to “3.3. hydrocarbon removal / transformation ”
We believe that these changes have made the manuscript clearer and with a better flow.

#Table 1. It is very difficult to understand the quantification data of the biodegradation of hydrocarbons. A simple way to represent these data is through three histograms in which, for each condition (ctr, oil and BS), the quantity of hydrocarbon at time 0 and after treatment at different temperatures are reported.

The authors understand the point raised by the reviewer in relation to difficulty in understanding hydrocarbon biodegradation quantification data. However, we believe that important information would be lost with another form of graphical representation. To solve it, we made small edits to the table and some information was clarified that presents the results.

#Pag. 12 Line 377. In the discussion, the Authors should consider that protists are component of the soil microbiome (see for instance https://doi.org/10.1093/femsre/fuy006) and some of these are able to live in extreme environments (see for instance https://doi.org/10.1016/j.ejop.2020.125720). It would be interesting to evaluate both the role of bacteria and that of protists in the decontamination of Antarctic soils.

The authors recognize the importance and the ability of protists to inhabit the soil microbiome and their ability to live in extreme environments. However, this is not the focus and objective of this article and certainly other research groups will be better able to clarify and develop research related to this taxonomic group.

#The Authors used a lot of abbreviations that make the text difficult to read and understand.

The authors searched for these terms and edited what could make understanding difficult. In addition, we hired an English editing service to improve the fluidity of the text and correct any errors. The text is now much better and to the point.

Minor comments:

#Title: amendeded should be amended.

Thank you. The typo was corrected 

#Pag. 2 Line 96. Please clarify how the Authors determine the physicochemical composition of the soil.

The methodology was referred now.

#Pag.3 Line 124. Please clarify why in the freeze and thaw cycles the Authors alternated the temperature every 15 days. Is it a standard protocol?

The authors alternated the temperature in this period according to the logistics and the time available to carry out the experiment and analysis. This period was chosen in an attempt to reconcile the time available with the best possible experiment design.

#Pag 3. Line 127 It is unclear the amount of soil the Authors used. Please clarify.

The amount of soil is specified at page 3 line 110: “50 g of soil in glass jars (100 ml)

#Pag.5 Line 210. Please add city and state.

This request was attended in line 210 by including the city.

#Figure 1. Figure resolution is too low.

The quality of this figure has been improved

#Figure 4. This figure is incomplete.

The authors do not understand this request because in their document this figure is complete. It was probably an error loading the first version. Either way, the figure has been improved and loaded correctly in the revised version.

We would like to thank you wholeheartedly for the reviewer's suggestions. The new revised version is much better now.

Round 2

Reviewer 1 Report

The revised paper reads better that the former. Howerver, there are still same issues:

L27 - don't use the word succession, replace with "response"

L33 - bacterial taxa in cursive 

L99 - nitrogen? what form?

L100 - phosphorous? what form?

L126 - according to this you would have 5 points of analysis. Why were these data not shown?

Table 1 - I still don't know what a, b, c means

Figure 4 - This graph was done in Excel not R ggplot2 as the authors claim. It has no scale, there are spelling mistakes in taxon names, 4B - genera and families? It the sum of percentages in each group in the B side all equal?

Author Response

Answers to Reviewer 1 (round 2)

L27 - don't use the word succession, replace with "response"

The word was replaced

L33 - bacterial taxa in cursive 

Corrected in the text. Thank you

L99 - nitrogen? what form?

Total nitrogen was the form evaluated. This information was included in line 99

L100 - phosphorous? what form?

Total soil available phosphorus was the form evaluated. This information was included in line 100

L126 - according to this you would have 5 points of analysis. Why were these data not shown?

Based on the ARISA result (figure 1) we realized that the intermediate collection points (samples) did not show any specific or variables related grouping. However, when analyzing initial and terminal samples (t0 and t14), the effects were clearly observed. Thus, aiming to consolidate the results and focus on the main results we decided to continue the downstream analyses by using only the samples which would better explain our questions.

The above explanation is already in the manuscript and can be found between lines 245 and 251.

Table 1 - I still don't know what a, b, c means

The legend of table 1 was edited to better explain the meaning of the superscripted lattes. For your convenience we also explain bellow:

Different samples with the same lattes (eg. Two samples with a superscript “a” indicate that the values found for both samples are not significantly different between each other)

Figure 4 - This graph was done in Excel not R ggplot2 as the authors claim. It has no scale, there are spelling mistakes in taxon names, 4B - genera and families? It the sum of percentages in each group in the B side all equal?

The authors claimed that PCA plot was generated in R software by using ggplot2 package among others, as explained in lines 221-223.

Figure 4 was reformulated to include the scale, to correct the spelling mistakes and also the sum of all groups in graph A and B.

Ps: Figure 4B was renamed to Operational taxonomic unity (OTUs) once in some cases was not possible to identify the genus level and upper levels are being represented.

We wish to thank the referee once again.

Reviewer 3 Report

Dear Editor

In the revised manuscript, the authors have been responded properly to all of comments to add descriptions and make corrections.

Minor comments:

Figure 1. The figure is incomplete, the green line is cut from the edge.

Pag.18. Supplementary Materials: Page: 18. Page 18 should be deleted.

Author Response

Answers to Reviewer 3 (round 2)

Figure 1. The figure is incomplete, the green line is cut from the edge.

The figure 1 was generated by the software in that form, it is not cut. Probably the software generated the figure in that way because the edges missing does not represent any additional or important importation to make the picture more comprehensive, there is no sample hidden. The black lines forming the square is limiting the entire figure.

As an example, the authors send below a link showing the same situation from another analysis:

http://www.lingyuetop.com/soft/pml/primer/index.htm

Pag.18. Supplementary Materials: Page: 18. Page 18 should be deleted.

This information was not present in our final revised version. It might be corrected during formatting the layout.

Thank you once more.